# Applications of maximum matching by using bipolar fuzzy incidence graphs

**Fahad Ur Rehman, Tabasam Rashid, Muhammad Tanveer Hussain** * 

Department of Mathematics, University of Management and Technology (UMT), Lahore, Pakistan

* tanveerhussain@umt.edu.pk

## Abstract

The extension of bipolar fuzzy graph is bipolar fuzzy incidence graph (*BFIG*) which gives the information regarding the effect of vertices on the edges. In this paper, the concept of matching in bipartite *BFIG* and also for *BFIG* is introduced. Some results and theorems of fuzzy graphs are also extended in *BFIGs*. The number of operations in *BFIGs* such as augmenting paths, matching principal numbers, relation between these principal numbers and maximum matching principal numbers are being investigated which are helpful in the selection of maximum most allied applicants for the job and also to get the maximum outcome with minimum loss (due to any controversial issues among the employees of a company). Some characteristics of maximum matching principal numbers in *BFIG* are explained which are helpful for solving the vertex and incidence pair fuzzy maximization problems. Lastly, obtained maximum matching principal numbers by using the matching concept to prove its applicability and effectiveness for the applications in bipartite *BFIG* and also for the *BFIG*.

**Data Availability Statement:** All relevant data are within the paper.

**Funding:** The author(s) received no specific funding for this work.

## 1 Introduction

A graph is more suitable to explain any kind of information along with the mutual relationship between different types of objects. The relationship between different entities are represented in terms of edges while entities do represent vertices. Zadeh was the first one who introduced the theory of fuzzy sets (*FSs*), which provides us the grade of membership of an object [1]. This theory opened an energetic area of research in various disciplines in the fields of automata, medical sciences, computer networking, statistics, social sciences and its various subbranches and disciplines, management sciences, engineering and graph theory etc. In this way Zadeh was the one who paved the way for Rosenfeld who introduced the fuzzy graph (*FG*) theory [2]. Rosenfeld was the one who presented several graph theoretical ideas, for example path, cycle and connectedness. *FG* is used when there is an inadequacy in the explanation and justification to various objects or entities and it was of great help to researcher. Mordeson put forth the concept regarding *FGs* and determined basic properties of it [3].

*FGs* are unable to give the detailed information about the impact of vertices on edges. This shortage in *FGs* was the basic problem which is covered by fuzzy incidence graphs (*FIGs*). The concept of *FIG* was put forth by Dinesh [4]. Different concepts with regard to the connectivity were put forth by Moderson and Mathew [5]. They introduced various structural properties

**Competing interests:** The authors have declared that no competing interests exist.

and establish the prevalence of a strong path between any of the node are pair of a *FIG*. Inter connectivity between index and wiener index with regard to the *FIGs* was put forth by Fang et al. [6].

Zhang introduced the concept of bipolar fuzzy sets (*BFSs*) [7]. The membership grade in the extension of *FS* to *BFS* is [-1,1]. An element has 0 grade in *BFS* if it has zero role on the resultant property. In such a way the membership degree of an element would be(0, 1] which will explain its properties to some extent. If membership grade of an element is [−1, 0) which tells that its marginal pleases the implicit counter property. The idea of the symbolization of bipolar fuzzy graphs (*BFGs*) along with the matrices in *FGs*, regular and irregular *BFGs*, hyper *BFGs* and antipodal *BFGs* along with their various applications, properties and significance was explained by Akram et al. [8–14]. Mohanta et al. gave a study of m-polar neutrosophic graph with applications [15]. Xiao et al. gave the study on regular picture fuzzy graph with applications in communication networks [16].

*FG* gives only positive membership values of vertices and edges whereas *FIGs* gives the positive membership values of vertices, edges and incidence pairs. *BFG* are able to give positive and negative membership values of vertices and edges. *FGs*, *FIG* and *BFGs* are unable to give the detailed information about the impact of vertices on edges. This shortage in *BFGs* was the basic problem which is covered by *BFIGs*. *BFGs* are able to give positive and negative membership values of vertices and edges whereas *BFIGs* are able to give positive and negative membership values of vertices, edges and incidence pairs. The concept of *BFIG* was put forth by Gong and Hua [17]. There are multiple reasons to introduce the concept of matching in bipartite *BFIG* and for *BFIG*. Let us consider an example to understand the concept of *BFIG*, if nodes reflects distinct companies and edges are the roads which connects these companies, then an *BFG* will give us the information of traffic between these companies. The company which have more number of employees will have the foremost infrastructure in the company. Hence, if $C_1$ and $C_2$ be two companies and $C_1C_2$ is a road between these companies, then $(C_1, C_1C_2)$ will be the incline system from the the company $C_1$ using the road $C_1C_2$ to the company $C_2$. Similarly, $(C_2, C_1C_2)$ will be the incline system from the company $C_2$ using the road $C_1C_2$ to the company $C_1$. Both $C_1$ and $C_2$ have the impact of 1 on $C_1C_2$ in un-weighted graphs. But, the impact of $C_1$ on $C_1C_2$ will be $(C_1, C_1C_2)$ is 1 whereas $(C_2, C_1C_2)$ is 0 in a directed graph. This is the main concept of *BFIG*.

Matching is important area in the graph as well as in the *FG* theory. It was Shen and Tsai who introduced the concept of optimal graph matching approach for solving the task assignment problem [18]. The concept of matching in *FGs* was introduced by Ramakrishnan and Vaidyanathan [19]. Later on, Mohan and Gupta further worked and gave the Graph matching algorithm for task assignment problem [20]. Matching numbers in fuzzy graphs are explained by Khalili et al. [21]. Our first objective is to find out maximum matching principal numbers in bipartite *BFIG* and for *BFIG* which are helpful to reflect the selection maximum applicants and their maximum working with minimum loss due to some controversial issues. Besides of this, some of the characteristics of the matching as well as bounds in bipartite *BFIGs* and *BFIG* have also been discussed. By using related examples, a detailed study has been carried out in the fields of matching number for the *BFIGs*.

Section 2 gives some preliminary definitions which are helpful to understand the next sections of the article. Section 3 contains some definitions, examples, results and theorems related to the concept of matching in *BFIG*. Section 4 gives mathematical model for obtaining *MMVBFIN* and *MMBFIN* for bipartite *BFIG* and *BFIG*. Section 5 contains comparative analysis is discussed for matching in bipartite *BFIG* and *BFIG*. Lastly, conclusions and prospects are explained in section 6.

## 2 Bipolar fuzzy incidence graph

This segment consists of some basic definitions including *FS*, *BFS*, *FG*, incidence graph (*IG*), *FIG*, *BFG*, *BFIG*, complete bipolar fuzzy incidence graph (*CBFIG*), matching, some concepts related to matching in classical theory and some examples. In this article, $V$, $E = V \times V$ and $I = V \times E$ represents the set of vertices, set of edges and set of incidence pairs, respectively. Let $G = (V, E)$ be a crisp graph. A set $\dot{M}$ of pairwise non-adjacent edges is known as matching. A matching $\dot{M}$ is known to be perfect matching if it covers all the vertices of the crisp graph $G$ and if a matching $\dot{M}$ covers maximum vertices then it is known as maximum matching. A crisp graph $G$ is said to be nearly perfect matching if only one vertex is unmatched. The number of edges in a maximum matching is known as the matching number and is denoted by $\alpha(\dot{M})$. A track in which edges are alternating in $\dot{M}$ and $E - \dot{M}$ is known as $\dot{M}$-alternating track and if neither its starting and nor its final vertex is covered by $\dot{M}$ then, it is known as $\dot{M}$-augmented track.

**Definition 2.1:** [1] Let $V$ be the *FS* from the universal set $U$ is defined as $V = \{(\chi_u, \nu_v(\chi_u)) : \nu_v(\chi_u) \in [0, 1], \chi_u \in U\}$.

**Definition 2.2:** [1] Let $V$ be the any nonempty set from the universal set $U$, a mapping $U : V \to [0, 1]$ is known as fuzzy subset.

**Definition 2.3:** [2] Let $\nu$ be the fuzzy subset of the set $V$ and $E$ be the fuzzy subset of $V \times V$. A *FG* $\hat{G} = (V, E)$ is a pair, such that $E(\nu_i, \nu_j) \leq \min(\nu(\nu_i), \nu(\nu_j)), \forall \nu_i, \nu_j \in V$.

**Definition 2.4:** [7] Let $U$ be a universal set. A *BFS* $B$ on $U$ is defined as $B = \{(\chi_u, \nu^p(\chi_u), \nu^n(\chi_u)) : \nu_1^p(\chi_u) \in [0, 1], \nu_1^n(\chi_u) \in [-1, 0], \chi_u \in U\}$.

**Definition 2.5:** [8] Let $\bar{G} = (V, E)$ be the *BFG* of *FG* $\hat{G} = (V, E)$ with the given conditions:

(a) $V = \{\nu_1, \nu_2, \nu_3, ..., \nu_n\} \ni \nu_1^p : U \to [0, 1]$ and $\nu_1^n : U \to [-1, 0]$,

(b) $E \subseteq V \times V$, $\nu_2^p : U \to [0, 1]$ and $\nu_2^n : U \to [-1, 0]$, such that

$$\nu_2^p(\nu_i, \nu_j) \leq \min(\nu_1^p(\nu_i), \nu_1^p(\nu_j)),$$
$$\nu_2^n(\nu_i, \nu_j) \geq \max(\nu_1^n(\nu_i), \nu_1^n(\nu_j)), \forall \nu_i, \nu_j \in E.$$

**Definition 2.6:** [4] Let $\bar{G} = (V, E, I)$ be an *IG* of a crisp graph $G = (V, E)$. Then $\tilde{G} = (V^*, E^*, I^*)$ be the *FIG* of *IG* $\bar{G}$, where $V^*$, $E^*$ and $I^*$ are the fuzzy subsets of $V$, $V \times V$ and $V \times E$ respectively, such that $I^*(\nu_i, \nu_i\nu_j) \leq \min(V^*(\nu_i), E^*(\nu_i\nu_j))$.

In Fig 1, the members of $I$ are $(q_0, q_0q_1)$, $(q_1, q_0q_1)$, $(q_1, q_1q_2)$ and $(q_2, q_1q_2)$.

**Definition 2.7:** [17] Let $\dot{G} = (V, E, I)$ be the *BFIG* of *FG* $\hat{G} = (V, E)$ with the given conditions:

(a) $V = \{\nu_1, \nu_2, \nu_3, ..., \nu_n\} \ni \nu_1^p : U \to [0, 1]$ and $\nu_1^n : U \to [-1, 0]$,

(b) $E \subseteq V \times V$, $\nu_2^p : U \to [0, 1]$ and $\nu_2^n : U \to [-1, 0]$,

(C) $I \subseteq V \times E$, $\nu_3^p : U \to [0, 1]$ and $\nu_3^n : U \to [-1, 0]$,

such that
$$\nu_3^p(\nu_i, \nu_i\nu_j) \leq \min(\nu_1^p(\nu_i), \nu_2^p(\nu_i\nu_j)),$$
$$\nu_3^n(\nu_i, \nu_i\nu_j) \geq \max(\nu_1^n(\nu_i), \nu_2^n(\nu_i\nu_j)), \forall \nu_i, \nu_j \in V \text{ and } \nu_i\nu_j \in E.$$

**Definition 2.8:** [17] Let $\dot{G} = (V, E, I)$ be the *BFIG*, then it is known as *CBFIG* if it satisfies the following conditions
$$\nu_3^p(\nu_i, \nu_i\nu_j) = \min(\nu_1^p(\nu_i), \nu_2^p(\nu_i\nu_j)),$$
$$\nu_3^n(\nu_i, \nu_i\nu_j) = \max(\nu_1^n(\nu_i), \nu_2^n(\nu_i\nu_j)), \forall(\nu_i, \nu_i\nu_j) \in I.$$

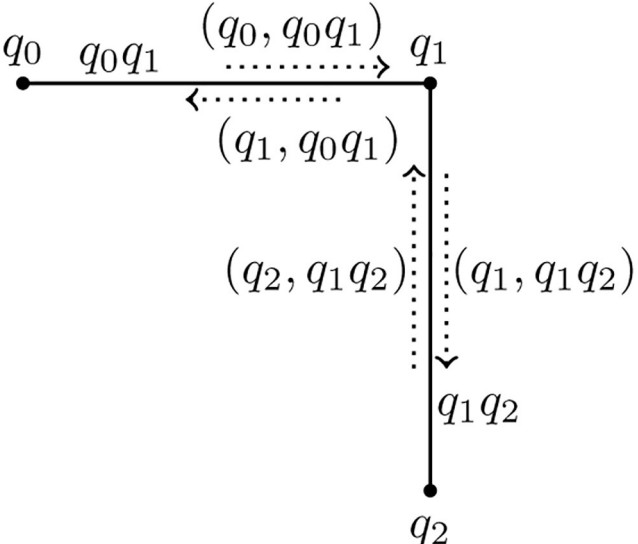

**Fig 1. An incidence graph $\dot{G}$.**

## 3 Matching in bipolar fuzzy incidence graph

This segment consists of some definitions like, support of *BFIG*, degree of vertices, degree of edges and incidence pairs in *BFIGs*, path, strength, strength of connectedness, matching, matching principal numbers, maximum matching principal numbers, some examples and theorems.

**Definition 3.1:** Let $\dot{G} = (V, E, I)$ be the *BFIG*, then the support of *BFIG* is denoted by $\dot{G} = (V^*, E^*, I^*)$ and is defined as

- $V^* = \{v_i \in V : v_1^p(v_i) > 0, v_1^n(v_i) > 0\}$,

- $E^* = \{v_i v_j \in V \times V : v_2^p(v_i v_j) > 0, v_2^n(v_i v_j) > 0\}$,

- $I^* = \{(v_i, v_i v_j) \in V \times E : v_3^p(v_i, v_i v_j) > 0, v_3^n(v_i, v_i v_j) > 0\}$.

**Definition 3.2:** Let $\dot{G} = (V, E, I)$ be the *BFIG*.

- Two vertices $v_0$ and $v_1$ are said to be connected if there exist a path from $v_0$ to $v_1$ such that $v_0$, $(v_0, v_0 v_1)$, $v_0 v_1$, $(v_1, v_1 v_0)$, $v_1$.

- Vertex $v_0$ and an edge $v_0 v_1$ are said to be connected if there exist a path such that $v_0$, $(v_0, v_0 v_1)$, $v_0 v_1$ between them.

**Definition 3.3:** Let $\dot{G} = (V^*, E^*, I^*)$ be the *BFIG*, then

- The degree of any vertex $v_i \in V^*$ in $\dot{G}$ is defined as $deg(v_i) = \sum_{v_j \in V^*, v_i \neq v_j} I(v_i, v_i v_j)$.

- The degree of any edge $E(v_i v_j) \in E^*$ in $\dot{G}$ is defined as $deg(v_i v_j) = \Sigma_{vk} \in VE(v_i v_k) + \Sigma_{vk} \in VE(v_j v_k) - 2E(v_i v_j)$.

- The degree of any incidence pair $I(v_i, v_i v_j) \in I^*$ in $\dot{G}$ is defined as $deg(v_i, v_i v_j) = \Sigma_{vk} \in VI(v_i, v_i v_k) + \Sigma_{vk} \in VI(v_j, v_j v_k) - 2I(v_i, v_i v_j)$.

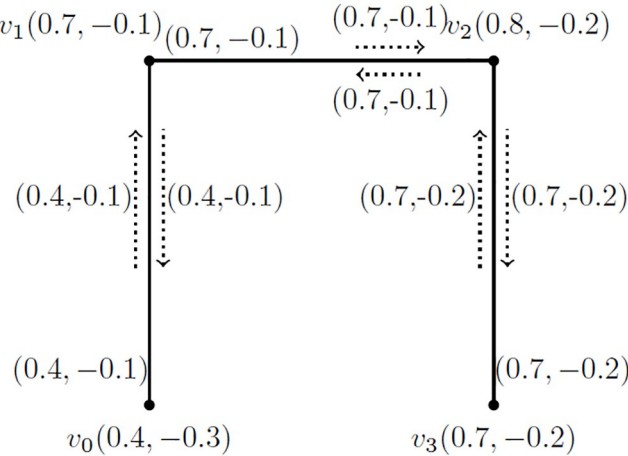

**Fig 2. An *BFIG* $\dot{G}$.**

**Example 3.4:** Consider a *BFIG*($\dot{G}$) as shown in Fig 2. We are going to calculate the degree of vertex and the incidence pair as well. The degree of distinct vertices is given as:

$deg(v_0) = (0.4, -0.1)$, $deg(v_1) = (1.1, -0.2)$, $deg(v_2) = (1.4, -0.3)$ and $deg(v_3) = (0.7, -0.2)$.

Similarly, the degree of distinct incidence pairs is given as:

$deg(v_0, v_0v_1) = (1.5, -0.3) - (0.4, -0.1) = (1.1, -0.2)$, $deg(v_1, v_1v_2) = (1.8, -0.4)$ and $deg(v_2, v_2v_3) = (1.4, -0.3)$.

**Definition 3.5:** The strength of connectedness between $v_i, v_j \in V(\dot{G})$ in the *BFIG* is denoted by $CONN_{\dot{G}}I(v_i, v_iv_j) = (CONN_G^p I(v_i, v_iv_j), CONN_G^n I(v_i, v_iv_j))$, where $CONN_G^p I(v_i, v_iv_j)$ and $CONN_G^n I(v_i, v_iv_j)$ are the maximum and the minimum of the strengths of all the paths between $v_i$ and $v_j$, respectively.

Throughout this article, strength of path is denoted by $S(P)$, strength of connectedness $CONN_{\dot{G}}(v_i, v_iv_j)$ will be represented by $I^\infty(v_i, v_iv_j)$. $\dot{M}$ is a matching of $\dot{G} = (V^*, E^*, I^*)$ with set of vertices, edges and incidence pair $V(\dot{M})$, $E(\dot{M})$ and $I(\dot{M})$ respectively. A collection of all matchings in $\dot{G}$ is denoted by $\dot{M}(\dot{G})$. A matching in $\dot{G}$ is known to be covering matching if $V = V(\dot{M})$.

**Definition 3.6:** Let $\dot{G} = (V^*, E^*, I^*)$ be the *BFIG* and its subgraph $H = (V', E', I')$ is known as matching in $\dot{G}$ if exactly single $v \in V'$ can be obtained $\forall u \in V'$ for which $v \neq u$ and $\mu_M(uv) \geq 0$.

**Example 3.7:** Consider a *BFIG* as given in Fig 3 with one possible matching. In this *BFIG*, we have:

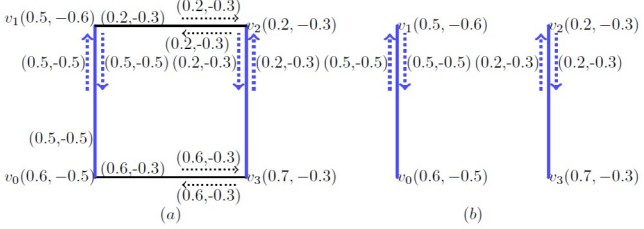

**Fig 3. A *BFIG* with a possible matching $\dot{M}$.**

$V(\dot{M}) = \{v_0, v_1, v_2, v_3\}$, $E(\dot{M}) = \{(v_0v_1), (v_2v_3)\}$ and $I(\dot{M}) = \{(v_0, v_0v_1), (v_2, v_2v_3)\}$.

**Corollary 3.8:** Let $\dot{G} = (V^*, E^*, I^*)$ be the *BFIG*. Any matching in $\dot{G}$ is induced by a matching in $G$.

**Proof** As a matching is taken as the set of triples like $\langle \ldots, v_i e_j v_k, \ldots \rangle$ and we must mention the vertices and incidence pair specifically. So, a matching $M$ as presented in Fig 3 and can be written as $\langle (v_0, v_0v_1), (v_2, v_2v_3) \rangle$.

**Proposition 3.9:** Let $\dot{G} = (V^*, E^*, I^*)$ be the *BFIG*. If $\dot{M}$ is a matching in $\dot{G}$, then $I^\infty(v_i, v_iv_j) = I(v_i, v_iv_j)$, for all $v_i, v_j \in V(\dot{M})$.

**Proof** Let $v_i, v_j \in V(\dot{M})$. If there is a path which connects $v_i$ and $v_j$, then this path is a single incidence pair $(v_i, v_iv_j)$ and $S(P) = I(v_i, v_iv_j)$, otherwise we have $S(P) = I(v_i, v_iv_j) = 0$. So, $I^\infty I(v_i, v_iv_j) = I(v_i, v_iv_j)$ for each case.

**Theorem 3.10:** Let $\dot{G} = (V^*, E^*, I^*)$ be the *BFIG* containing a matching $\dot{M}$, then $deg(v_i) = deg(v_j) = (v_i, v_iv_j)$ and $deg(v_i, v_iv_j) = 0$ for every $v_iv_j \in \dot{M}$.

**Proof** As $\forall v_i \in V(\dot{M})$, there is only one $v_j \in V(\dot{M})$ is available such that $I(v_i, v_iv_j) > 0$, we get

$deg(v_i) = \sum_{v_k \in V(\dot{M}), v_k \neq v_i} I(v_i, v_iv_k) = \sum_{v_k = v_j} I(v_i, v_iv_k) = I(v_i, v_iv_j)$,

and

$degI(v_i, v_iv_j) = \sum_{v_k \in V(\dot{M})} I(v_i, v_iv_k) + \sum_{v_k \in V(\dot{M})} I(v_j, v_jv_k) - 2I(v_i, v_iv_j)$,

$degI(v_i, v_iv_j) = \sum_{v_k = v_j} I(v_i, v_iv_k) + \sum_{v_k = v_i} I(v_j, v_jv_k) - 2I(v_i, v_iv_j)$,

$degI(v_i, v_iv_j) = I(v_i, v_iv_j) + I(v_j, v_jv_i) - 2I(v_i, v_iv_j) = 0$.

**Definition 3.11:** Let $\dot{M}$ is a matching in BFIG $\dot{G} = (V^*, E^*, I^*)$. Then,

- The matching bipolar fuzzy incidence number of $\dot{M}$ can be described as,
  $\alpha_I(\dot{M}) = \sum_{I \in I(\dot{M})} I(v_i, v_iv_j)$.

- The matching edge bipolar fuzzy incidence number of $\dot{M}$ can be described as,
  $\alpha_E(\dot{M}) = \sum_{e \in E(\dot{M})} E(v_iv_j)$.

- The matching vertex bipolar fuzzy incidence number of $\dot{M}$ can be described as,
  $\alpha_V(\dot{M}) = \sum_{v_i \in E(\dot{M})} V(v_i)$.

- The matching crisp number of $\dot{M}$ can be described as, $\alpha_C(\dot{M}) = |I(\dot{M})|$.

We consider $\alpha_I(\dot{M})$, $\alpha_V(\dot{M})$ and $\alpha_C(\dot{M})$ as matching bipolar fuzzy incidence principal numbers (*MBFIPNs*) of $\dot{M}$.

**Example 3.12:** A *BFIG* with a possible matching is presented in Fig 3. The *MBFIPNs* are obtained as,

$\alpha_V(\dot{M}) = (0.7, -0.8)$, $\alpha_I(\dot{M}) = (2, -1.7)$ and $\alpha_C(\dot{M}) = 2$.

**Definition 3.13:** Let $\dot{M}$ is a matching in BFIG $\dot{G} = (V^*, E^*, I^*)$. Then,

- The maximum matching bipolar fuzzy incidence number of $\dot{G}$ can be described as:
  $\alpha_I^{\max}(\dot{M}) = \max\{\alpha_I(\dot{M}) : M \in \dot{M}(\dot{G})\}$.

- The maximum matching edge bipolar fuzzy incidence number of $\dot{G}$ can be described as:
  $\alpha_E^{\max}(\dot{M}) = \max\{\alpha_E(\dot{M}) : M \in \dot{M}(\dot{G})\}$.

- The maximum matching vertex bipolar fuzzy incidence number of $\dot{G}$ can be described as:
  $\alpha_V^{\max}(\dot{M}) = \max\{\alpha_V(\dot{M}) : M \in \dot{M}(\dot{G})\}$.

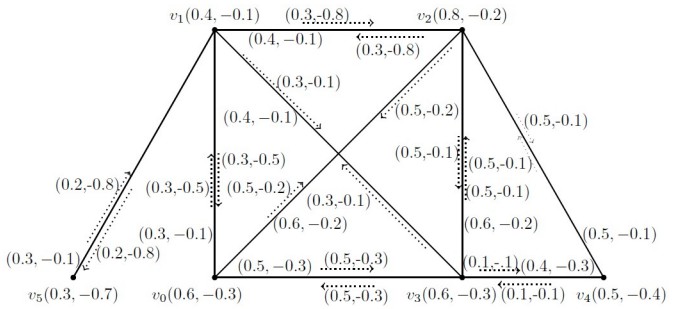

**Fig 4. A *BFIG* ($\dot{G}$).**

- The maximum matching crisp number of $\dot{G}$ can be described as:
$$\alpha_C^{\max}(\dot{M}) = \max\{\alpha_C(\dot{M}) : \dot{M} \in \dot{M}(\dot{G})\}.$$

We consider $\alpha_I^{\max}(\dot{M})$, $\alpha_V^{\max}(\dot{M})$ and $\alpha_C^{\max}(\dot{M})$ as *MMBFIN*, *MMVBFIN* and *MMCN*.

In classical graph theory, a lot of matchings with same *MMCN* can be found but in fuzzy sense, we can differentiate them in terms of fuzzy values.

**Example 3.14:** Consider a *BFIG* $\dot{G} = (V^*, E^*, I^*)$ as given in Fig 4.

Now, we will find out all possible matchings, *MBFIPNs* and after that *MMBFIN*, *MMVBFIN* and *MMCN* for Fig 4 as presented in Table 1.

Now, it is easy to calculate the following numbers: $\alpha_I^{\max}(\dot{M}) = (1.2, -1.2)$, $\alpha_V^{\max}(\dot{M}) = (3.2, -2)$ and $\alpha_C^{\max}(\dot{M}) = 3$.

**Proposition 3.15:** Let $\dot{M}$ be a matching in *BFIG* $\dot{G} = (V^*, E^*, I^*)$. Then, $\forall \dot{M} \in \dot{M}(\dot{G})$, we have $\alpha_I(\dot{M}) < \alpha_E(\dot{M}) < \alpha_V(\dot{M})$.

**Proof** Let $\dot{M} \in \dot{M}(\dot{G})$. As $\dot{M}(\dot{G})$ is a *BFIG*,

$I(v_i, v_i v_j) \leq V(v_i) \wedge E(v_i v_j)$ and $E(v_i v_j) \leq V(v_i) \wedge V(v_j)$ for all $I = (v_i, v_i v_j) \in \dot{M}$. So, we have:

$$\alpha_I(\dot{M}) = \sum_{I(v_i, v_i v_j) \in \dot{M}} I(v_i, v_i v_j) < \sum_{E(v_i v_j) \in E(\dot{M})} E(v_i v_j) < \sum_{v_i \in V(\dot{M})} V(v_i) = \alpha_V(\dot{M}).$$

**Definition 3.16:** Let $\dot{G} = (V^*, E^*, I^*)$ be a *BFIG* containing a matching $\dot{M}$. A bipolar fuzzy $\dot{M}$-augmenting track in $\dot{G}$ is an $\dot{M}$-alternating track containing different nodes $v_o, v_1, v_2, \ldots v_n, v_{n+1}$. So, as a result:

- $I(v_{i-1}, v_{i-1} v_i) > 0$, where $i = 1, 2, 3, \ldots, n, n + 1$,

**Table 1. All possible matchings and *MBFIPNs* of *BFIG* for Fig 4.**

| Sr. No | Matching possibilities | $\alpha_I(M)$ | $\alpha_V(M)$ | $\alpha_C(M)$ |
|---|---|---|---|---|
| 1 | $\{I(v_1, v_1 v_5), I(v_2, v_2 v_3)\}$ | $(0.7, -0.9)$ | $(2.1, -1.3)$ | 2 |
| 2 | $\{I(v_0, v_0 v_1), I(v_2, v_2 v_3)\}$ | $(0.8, -0.6)$ | $(2.4, -0.9)$ | 2 |
| 3 | $\{I(v_1, v_1 v_2), I(v_0, v_0 v_3)\}$ | $(0.8, -1.1)$ | $(2.4, -0.9)$ | 2 |
| 4 | $\{I(v_1, v_1 v_3), I(v_0, v_0 v_2)\}$ | $(0.8, -0.3)$ | $(2.4, -0.9)$ | 2 |
| 5 | $\{I(v_3, v_3 v_4), I(v_0, v_0 v_2)\}$ | $(0.6, -0.3)$ | $(2.5, -1.2)$ | 2 |
| 6 | $\{I(v_2, v_2 v_4), I(v_0, v_0 v_3), I(v_1, v_1 v_5)\}$ | $(1.2, -1.2)$ | $(3.2, -2)$ | 3 |
| 7 | $\{I(v_1, v_1 v_3), I(v_0, v_0 v_2)\}$ | $(0.9, -0.2)$ | $(2.3, -1)$ | 2 |
| 8 | $\{I(v_2, v_2 v_4), I(v_0, v_0 v_1)\}$ | $(0.8, -0.6)$ | $(2.3, -1)$ | 2 |
| 9 | $\{I(v_0, v_0 v_1), I(v_3, v_3 v_4)\}$ | $(0.4, -0.6)$ | $(2.1, -1.1)$ | 2 |

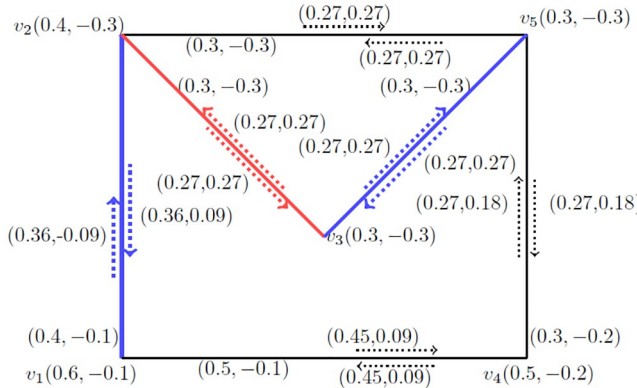

**Fig 5. *BFIG* for the comparing of *MBFIPNs* and $(P \oplus \dot{M})$.**

- $\{v_o, v_1, v_2, \ldots v_n, v_{n+1}\} \subseteq V^*(\dot{M})$,

- Neither $v_0$ nor $v_{n+1}$ are in $V^*(\dot{M})$.

**Corollary 3.17:** Let $\dot{G} = (V^*, E^*, I^*)$ be a *BFIG* containing a bipolar fuzzy incidence $\dot{M}$-augmenting track $P$. Then, it is $\dot{M}$-augmenting track in crisp graph $G = (V, E)$.

**Proof** Let $\dot{G} = (V^*, E^*, I^*)$ be a *BFIG* containing a matching $\dot{M}$. $P$ be a bipolar fuzzy incidence $\dot{M}$-augmenting track and their symmetric difference is denoted by $\oplus$. As $P \oplus \dot{M}$ represents a collection of nonadjacent incidence pairs and $I(v_i, v_i v_j) > 0$ for all $I \in P \cap \dot{M}$, which shows that $P \oplus \dot{M}$ is a matching.

**Theorem 3.18:** Let $\dot{G} = (V^*, E^*, I^*)$ be a *BFIG* containing a matching $\dot{M}$. If $P$ is a bipolar fuzzy incidence $\dot{M}$-augmenting track then, $\alpha_V^*(P \oplus \dot{M}) > \alpha_V^*(\dot{M})$.

**Proof** Let $P$ be a bipolar fuzzy $\dot{M}$-augmenting track, by using definition 3.16, we have
$V^*(P \oplus \dot{M}) = V^*(\dot{M}) \cup \{v_0, v_{n+1}\}$.
Now, by using definition 3.11, we have:
$\alpha_V^*(P \oplus \dot{M}) = \sum_{v_i \in V^*(P \oplus \dot{M})} V(v_i) + V(v_0) + V(v_{n+1})$,
$\alpha_V^*(P \oplus \dot{M}) = \alpha_V^*(\dot{M}) + V^*(v_0) + V^*(v_{n+1})$,
As a result, we get: $\alpha_V^*(P \oplus \dot{M}) > \alpha_V^*(\dot{M})$.

**Example 3.19:** Consider a *BFIG* $(\dot{G})$ as given in Fig 5. Now, matching $\dot{M}$ and the incidence pair in the augmented track between $v_1, v_5$ is $\langle (v_2, v_2 v_3) \rangle$. But in $P \oplus \dot{M}$, it is $\langle (v_1, v_1 v_2), (v_3, v_3 v_5) \rangle$ as presented in Table 2 for Fig 5.

**Theorem 3.20:** Let $\dot{G} = (V^*, E^*, I^*)$ be a *BFIG* containing a matching $\dot{M}$ with *MMVBFIN*. Then, $\dot{M}$ has *MMCN*.

**Table 2. Comparing the *MBFIPNs* and $P \oplus \dot{M}$ of *BFIG* as given in Fig 5.**

| Sr. No | MBFIPNs | M | $P \oplus M$ |
|---|---|---|---|
| 1 | $\alpha_V$ | (0.3, −0.3) | (0.7, −0.4) |
| 2 | $\alpha_E$ | (0.7, −0.6) | (1.6, −1) |
| 3 | $\alpha_I$ | (0.7, −0.6) | (1.6, −1) |
| 4 | $\alpha_C$ | 1 | 2 |

**Proof** Let $\dot{G} = (V^*, E^*, I^*)$ be a *BFIG* containing a matching $\dot{M}$ with *MMVBFIN*. It is adequate to prove that $\dot{M}$ has foremost number of incidence pairs. As $\dot{M}$ is a matching in crisp graph $G$. Now, if there exist any $\dot{M}$-augmenting track $P$, then by using the symmetric difference $P \oplus \dot{M}$ increases the *MVBFIN* by using theorem 3.18. So, the condition of maximum incidence pairs holds. Hence, in a matching $M$ with *MMVBFIN* there exist *MMCN*. Therefore, according to the Berge's theorem, $\dot{M}$ has the foremost number of edges, if there is no $\dot{M}$-augmenting track.

**Remark 3.21:** As a bipolar fuzzy incidence covering matching (*BFICM*) includes all the vertices of *BFIG*, hence each bipolar fuzzy covering matching acknowledges *MMVBFIN*.

**Corollary 3.22:** Let $\dot{G} = (V^*, E^*, I^*)$ be a *BFIG* containing *BFICM* ($\dot{M}$), then $\dot{M}$ admits *MMVBFIN*.

**Proof** Let $\dot{G} = (V^*, E^*, I^*)$ be the *BFIG*. If there does not exist any $\dot{M}$-augmenting track then, there must exist at least one $\dot{M}$,. So, according to the Berge's theorem, if there does not exist any $\dot{M}$-augmenting track then, $\dot{M}$ has the foremost number of edges. Hence, it must admits *MMVBFIN*.

**Definition 3.23:** Let $\dot{G} = (V^*, E^*, I^*)$ be the *BFIG*. Then:

- Consider two arbitrary nodes $v_1, v_2 \in V^*$. $v_1$ is known as bipolar fuzzy incidence prior to $v_2$ if and only if $v_1^p(v_1) \leq v_1^p(v_2), v_1^n(v_1) \leq v_1^n(v_2)$ and $l(v_1) \leq l(v_2)$. It is denoted by $v_1 \prec v_2$.

- Let Let $\dot{G} = (V^*, E^*, I^*)$ be the *BFIG* with two matchings $\dot{M}^1$ and $\dot{M}^2$, for which $|V^*(\dot{M}^1)| = |V^*(\dot{M}^2)|$. Then, $\dot{M}^1$ is known as bipolar fuzzy incidence prior to $\dot{M}^2$ if and only if $\alpha_V^*(\dot{M}^1) < \alpha_V^*(\dot{M}^2)$.

- Consider $\{\dot{M}^i \mid 1 \leq i \leq n\}$ is the set which includes all the possible matchings in $\dot{G}$ with *MMCN*. A matching $\dot{M}^{\max}$ is known as bipolar fuzzy incidence strong vertex matching, if $\dot{M}^i \prec \dot{M}^{\max}$, where $i = 1, \ldots, n$.

**Proposition 3.24:** Let $\dot{G} = (V^*, E^*, I^*)$ be a *BFIG*. If $\dot{M}^{max}$ is a bipolar fuzzy incidence strong vertex subgraph in $\dot{G} = (V^*, E^*, I^*)$, then $\alpha_V(\dot{M}^{\max}) = \alpha_V^{\max}$.

**Proof** Consider $\dot{M}$ from $\{\dot{M}^i \mid 1 \leq i \leq n\}$. By using the theorem 3.20, any $\dot{M}^i$ admits the *MMCN*. So, by using the definition of bipolar fuzzy incidence strong vertex matching, we have $\alpha_V(\dot{M}^i) \leq \alpha_V(\dot{M}^{\max})$. Hence, $\alpha_V(\dot{M}^{\max}) = \alpha_V^{\max}$.

**Definition 3.25:** Let $\dot{G} = (V^*, E^*, I^*)$ be the *BFIG*. Then, it is called bipartite bipolar fuzzy incidence graph (*BBFIG*) if the set of vertices $V$ can be divided into two subsets $V_1$ and $V_2$ such that each edge either connects a vertex from $V_1$ to $V_2$ or a vertex from $V_2$ to $V_1$.

**Remark 3.26:** Every perfect matching of $\dot{G}$ is the spanning graph of $\dot{G}$.

Now, we are going to construct pseudo-fuzzy restrictions for the *BBFIG* which will be used in different methods for finding the matchings with *MBFIPNs*.

**Definition 3.27:** Let $\dot{G}$ be the *BBFIG*. The set of vertices $V$ is divided into two subsets $V_1$ and $V_2$ such that $V = V_1 \cup V_2$. We consider the pseudo bipolar fuzzy incidence restrictions for $\dot{G}^{V_1}$ as:

$$I(\dot{G}^{V_1}) = I(\dot{G}), E(\dot{G}^{V_1}) = E(\dot{G}), V(\dot{G}^{V_1}) = V(\dot{G}) = V \text{ and } v_{V_1}(v) = \begin{cases} v(v), & \text{if } v \in V_1 \\ l(v), & \text{if } v \in V_2 \end{cases}.$$

In the same way, we consider the pseudo bipolar fuzzy incidence restrictions for $\dot{G}^{V_2}$ as:

$$I(\dot{G}^{V_2}) = I(\dot{G}), E(\dot{G}^{V_2}) = E(\dot{G}), V(\dot{G}^{V_2}) = V(\dot{G}) = V \text{ and } v_{V_2}(v) = \begin{cases} v(v), & \text{if } v \in V_2 \\ l(v), & \text{if } v \in V_1 \end{cases}.$$

**Theorem 3.28:** Let $\dot{M}^{V_1}$ and $\dot{M}^{V_2}$ be the two matchings, respectively in the pseudo bipolar fuzzy incidence restrictions $\dot{G}^{V_1}$ and $\dot{G}^{V_2}$ of *BBFIG* $\dot{G}$ with $V = V_1 \cup V_2$ as the set of vertices. Then, there is a new matching $\dot{M} \subseteq \dot{M}^{V_1} \cup \dot{M}^{V_2}$, which matches all the vertices covered by $\dot{M}^{V_1}$ and $\dot{M}^{V_2}$.

**Proof** Let $\dot{G}$ be the *BBFIG*. Let $A \subseteq V_1$ and $B \subseteq V_2$. If $\dot{G}$ has a matching covering $A$ and matching covering $B$, then it has a matching covering $A \cup B$. Hence, if $\dot{M}^{V_1}$ and $\dot{M}^{V_2}$ be the two matchings, respectively in the pseudo fuzzy restrictions $\dot{G}^{V_1}$ and $\dot{G}^{V_2}$ of *BBFIG* $\dot{G}$ with $V = V_1 \cup V_2$ as the set of vertices. There is a new matching $\dot{M} \subseteq \dot{M}^{V_1} \cup \dot{M}^{V_2}$, which matches all the vertices covered by $\dot{M}^{V_1}$ and $\dot{M}^{V_2}$.

# 4 Mathematical model

In this section, we will discuss the method for obtaining *MMBFIN*. There are two objectives for achieving *MMBFIN*. First is to give the maximum jobs to the applicants where two factors are focused: (1) maximize positive membership value which reflects their maximum working efficiency of the applicants. (2) minimizing the negative membership value which reflects the bad performance due to controversial issues among them. Second objective is to maximize the working of the employees of a company. To achieve the first objective *BBFIG* is used whereas to achieve the second objective *BFIG* is used.

## 4.1 MMVBFIN problem in BBFIG

In this part, we are explaining the process to obtain the *MMVBFIN* in *BBFIG*. In this process the main points are;

*Step*—1 Arrange the vertices of $V_1$ and $V_2$ in ascending order.

*Step*—2 Let $v_1 \in V_1$ be a vertex having highest membership value which in matched with vertex from $V_2$ having highest membership value and obtain the matching $(\dot{M}_1)$ for the graph.

*Step*—3 Consider that matching and find another matching by taking symmetric difference of $\dot{M}_1$. Continue this process till there is no augmenting path or obtain the matching which is already obtained.

*Step*—4 Choose the strongest matching and obtain the *MMVBFIN*.

**Example 4.1:** Let $\dot{G}$ be the *BBFIG* as presented in Fig 6.

*Step*—1 by using this step we have arranged the vertices according to their membership values and presented in Fig 7.

*Step*—2 By using second step, we get $\dot{M}_1$ which is $\dot{M}_1 = \langle (u_1, u_1 v_3), (u_3, u_3 v_2), (u_2, u_2 v_0) \rangle$ and its augmenting path is represented by $(u_1, u_1 v_2), (u_2, u_2 v_1)$ as shown in Fig 8.

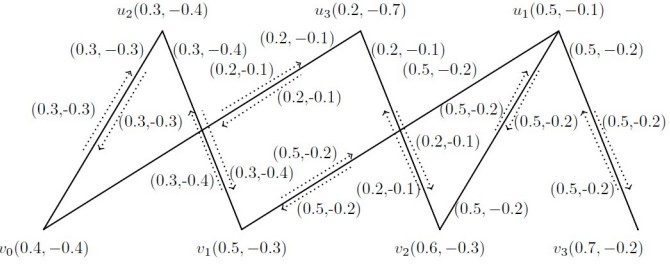

**Fig 6. An *BBFIG*.**

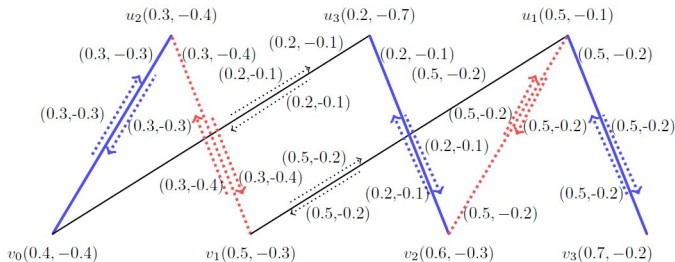

**Fig 7. An *BBFIG*.**

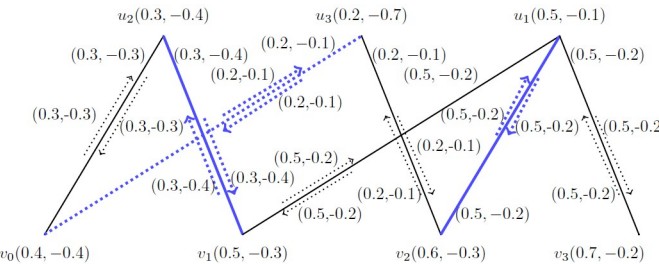

**Fig 8. An *BBFIG*.**

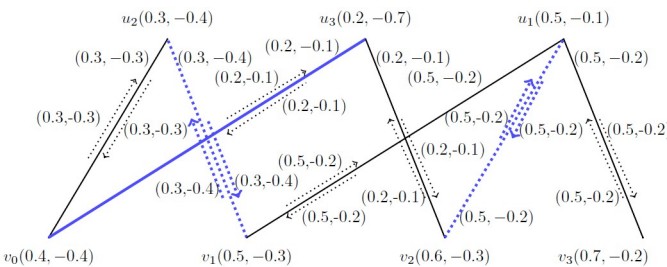

**Fig 9. An *BBFIG*.**

*Step*—3 By using third step, we get $\dot{M}_2$ which is $\dot{M}_2 = \langle (u_1, u_1 v_2), (u_2, u_2 v_1) \rangle$ and its augmenting path is represented by $(u_3, u_3 v_0)$. Now, obtaining again its augmenting path presented in Fig 9;

*Step*—3 By using third step, we get $\dot{M}_3$ which is $\dot{M}_3 = \langle (u_3, u_3 v_0) \rangle$ and its augmenting path is represented by $(u_1, u_1 v_2), (u_2, u_2 v_1)$ which is same as $\dot{M}_2$ as shown in Fig 10.

Hence, by using step-4, we get $\dot{M}_1$ as our final matching and *MMVBFIN* = (2.7, −2.1).

## 4.2 MMVBFIN problem in arbitrary BFIG

In this part, we are explaining the process to obtain the *MMVBFIN* in the arbitrary *BFIG*. In this process the main points are;

*Step*—1 Arrange the vertices such that $v_1$ is strongest vertex. $v_2$ is the vertex which is connected with $v_1$ and weaker than $v_1$.

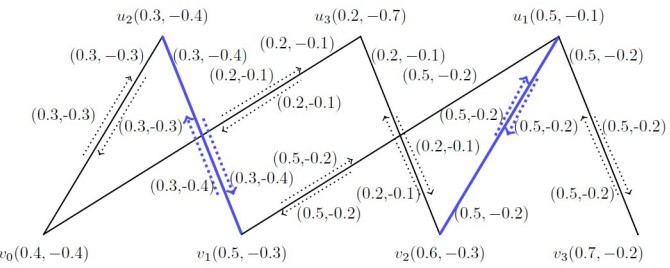

**Fig 10. An *BBFIG*.**

*Step—2* Consider an incidence pair, except $(v_1, v_1v_2)$ connected with $v_2$ which is our first matching. If, no such incidence pair is found then, start from $(v_1, v_1v_2)$.

*Step—3* Obtain the strong vertex augmenting path from $\dot{M}_1$ and continue this process untill there is no augmenting path.

*Step—4* Choose the maximum vertex matching and obtain *MMVBFIN*.

**Application** Let $\dot{G}$ be the *BFIG* as shown in Fig 11.

*Step–1* By using step one, $v_1$ is strongest vertex. $v_2$ is the vertex which is connected with $v_1$ and weaker than $v_1$. $v_3$ is weaker then $v_2$ and so on.

*Step—2* The incidence pair $\dot{M}_1 = (v_2, v_2v_3)$ is our first matching and the augmenting path is $p_1 = \langle (v_2, v_2v_1), (v_3, v_3v_4) \rangle$.

*Step—3* Taking $\dot{M}_2 = \langle (v_2, v_2v_1), (v_3, v_3v_4) \rangle$ is second matching and the augmenting path is $p_2 = \langle (v_1, v_1v_5), (v_4, v_4v_6), (v_2, v_2v_3) \rangle$.

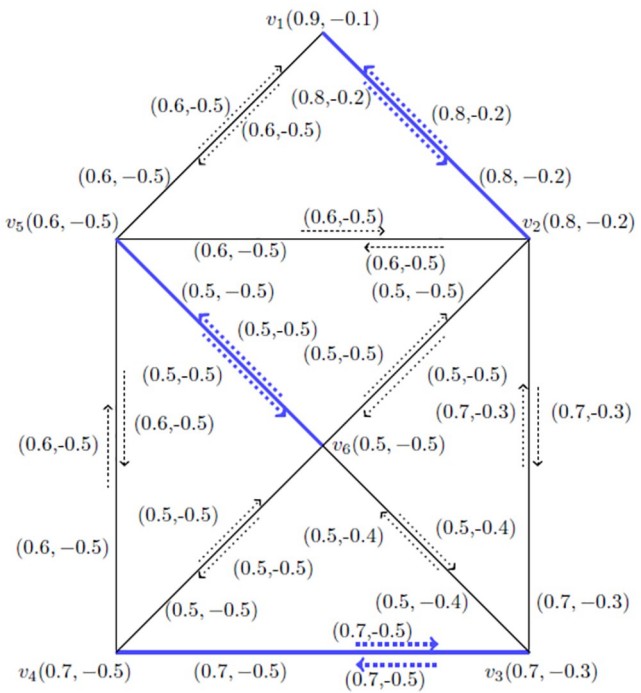

**Fig 11. An arbitrary *BFIG*.**

Continuing this process, $\dot{M}_3 = \langle (v_1, v_1v_5), (v_4, v_4v_6), (v_2, v_2v_3) \rangle$ is third matching and the augmenting path is $p_3 = \langle (v_2, v_2v_1), (v_5, v_5v_6), (v_3, v_3v_4) \rangle$.

Furthermore, $\dot{M}_4 = \langle (v_2, v_2v_1), (v_5, v_5v_6), (v_3, v_3v_4) \rangle$ is our last matching and its augmenting path is again same as $\dot{M}_3$.

*Step*—4 Now, $M_3$ and $M_4$ both have *MMVBFIN*. $\dot{M}_3 = \dot{M}_4 = \langle (v_2, v_2v_1), (v_5, v_5v_6), (v_3, v_3v_4) \rangle$ is maximum vertex matching and *MMVBFIN* = (4.2, −1.8) for Fig 11.

## 4.3 MMBFIN problem in BBFIG

The matching concept is used for *BBFIG* because it is considerably more complicated in *BFIG*. The question is weather the difficulty is reducible or not? By using the proposed method for the *BFIG*, the answer is "*yes*". Firstly, we are going to introduce the method for obtaining the *MMBFIN* in the *BBFIG*. The greatest membership value of the incidence pair in $\dot{G}$ is presented by $I_{\max}(\dot{G})$. In simple words:

$I_{\max}(\dot{G}) = \max I(v_i, v_iv_j)$ and $I_{\max}(v_i)\dot{G} = \max_{I(v_i, v_iv_j)} I(v_i, v_iv_j)$.

Let $\dot{G}$ be the *BBFIG* with $V = V_1 \cup V_2$. For any vertex $v \in V$, there exist one and only one adjacent incidence pair of $v$ which is present in the matching process. Taking $y : I \to \{0, 1\}$ as an incidence vector which reflects the presence or absence of an incidence pair in the matching. The *MMBFIN* problem in the *BBFIG* can be described as:

$z = (\max \Sigma y(I)I^p, \min \Sigma y(I)I^n)$

subject to: $\Sigma_{I = (vi, v_iv_j)} I = 1, \forall v \in V, I \in \{0, 1\}$.

**Example 4.2:** Let $\dot{G}$ be the *BBFIG* with $\dot{M}$. Consider there are three jobs and four applicants which are represented by $\{u_1, u_2, u_3\}$ and $\{v_0, v_1, v_2, v_3\}$. Our goal is to assign jobs to applicants and every job is assigned to at most one applicant such that maximum number of jobs will be filled by using the matching concept.

$z = (0.3(u_2, u_2v_0) + 0.3(u_2, u_2v_1) + 0.2(u_3, u_3v_0) + 0.2(u_3, u_3v_2) + 0.5(u_1, u_1v_1) + 0.5(u_1, u_1v_2) + 0.5(u_1, u_1v_3), -0.4(u_2, u_2v_0) - 0.4(u_2, u_2v_1) - 0.7(u_3, u_3v_0) - 0.7(u_3, u_3v_2) - 0.1(u_1, u_1v_1) - 0.1(u_1, u_1v_2) - 0.1(u_1, u_1v_3))$,

Subject to:

$(u_2, u_2v_0) + (u_2, u_2v_1) = 1$,

$(v_0, v_0u_2) + (v_0, v_ou_3) = 1$,

$(u_3, u_3v_0) + (u_3, u_3v_2) = 1$,

$(v_1, v_1u_2) + (v_1, v_1u_1) = 1$,

$(u_1, u_1v_1) + (u_1, u_1v_2) + (u_1, u_1v_3) = 1$.

$(v_2, v_2u_3) + (v_2, v_2u_1) = 1$,

$(v_3, v_3u_1) = 1$,

The above system of linear equations is solved by using the simplex method and got *MMBFIN* $= \alpha_I^{\max} = (1, -0.6)$ for Fig 12.

By using the matching process $\{v_0, v_2, v_3\}$ got the job. For the job $u_1$ the applicant $v_3$ is selected because the negative membership value of $v_3$ is less then $v_1, v_2$. Similarly, other applicants are selected. By using this concept maximum applicants are selected as all the vacancies are filled.

## 4.4 MMBFIN problem in arbitrary BFIG

In this part, we are explaining the process to obtain the *MMBFIN* in the arbitrary *BFIG*. In this process the main points are;

*Step*—1 Let the matching $\dot{M}$ be empty by default for $(\dot{G})$.

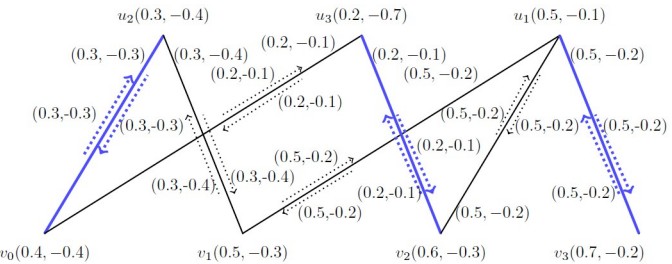

**Fig 12. An *BFIG* with $\dot{M}$.**

*Step*—2 The incidence pair $I$ is known as elected incidence pair for the subgraph $H^i$. The criteria of selecting the vertices for obtaining the *MMBFIN* of $\dot{G}$ is $v(vi) \geq \frac{1}{2}I_{\max}(v_i)\dot{G}$ and $v(vj) \geq \frac{1}{2}I_{\max}(v_j)\dot{G}$. If there are more then one incidence pairs which fulfills this criteria then select the incidence pair having maximum membership value.

*Step*—3 Select the alternating incidence pair from adjacent incidence pairs. If there are more then one incidence pairs then, make a matching as $\dot{M}_i$ for every incidence pair.

*Step*—4 Repeat the same process for all the possible $\dot{M}$ of $\dot{G}$.

*Step*—5 Select the matching having $\alpha_I^{\max}(\dot{M}(\dot{G}))$ which gives *MMBFIN* and also shows the maximum working of the employees in a company.

**Application** Consider a department have 6 members. The 6 members are our vertices. We give them membership values according to their individual performance. There edge values are defined as their work performance with other member as a group. The positive membership value of incidence pair value defines the working efficiency of two employees in a company and negative membership value defines their loss possibility in the working due to controversial issues among the employees as a group. By the matching process, we will get the best match of partners.

Let $\dot{G}$ be the *BFIG* and our destination is to obtain the matching with *MMBFIN* of $\dot{G}$.

*Step*—1 Let the matching $\dot{M}$ be empty by default for a *BFIG* ($\dot{G}$).

*Step*—2 The incidence pair $(v_1, v_1v_2)$ is elected incidence pair by using the criteria of selection of the vertices as mentioned above.

*Step*—3 The matching $\dot{M}_1 = \langle (v_1, v_1v_2), (v_3, v_3v_4), (v_5, v_5v_6) \rangle$ is obtained.

*Step*—4 By using this step, we obtained $\dot{M}_2 = \langle (v_1, v_1v_2), (v_3, v_3v_6), (v_4, v_4v_5) \rangle$ and $\dot{M}_3 = \langle (v_1, v_1v_2), (v_4, v_4v_6) \rangle$.

*Step*—5 Lastly, the $M_1$ is selected as shown in Fig 13 and *MMBFIN* is computed as $\Rightarrow \alpha_I^{\max}(\dot{M}(\dot{G})) = (2, -1.2)$.

By using this process, the best partners are selected which gives us maximum working efficiency and the chances of loss due to any controversial issues among the employees in a company are also minimized.

## 5 Comparative analysis

In Fig 7, there were three jobs $\{u_1, u_2, u_3\}$ and four applicants $\{v_0, v_1, v_2, v_3\}$. Our target is to give maximum jobs to the applicants in order to get maximum working efficiency and minimizing the low performance due to controversial issues among the applicants. By using the matching concept, we obtained *MMVBFIN* = $(2.7, -2.1)$ and got a matching $\dot{M}_1 = \langle (u_1, u_1v_3), (u_3, u_3v_2), (u_2, u_2v_0) \rangle$ for *BBFIG* by using vertices. Now, by using Fig 12 we

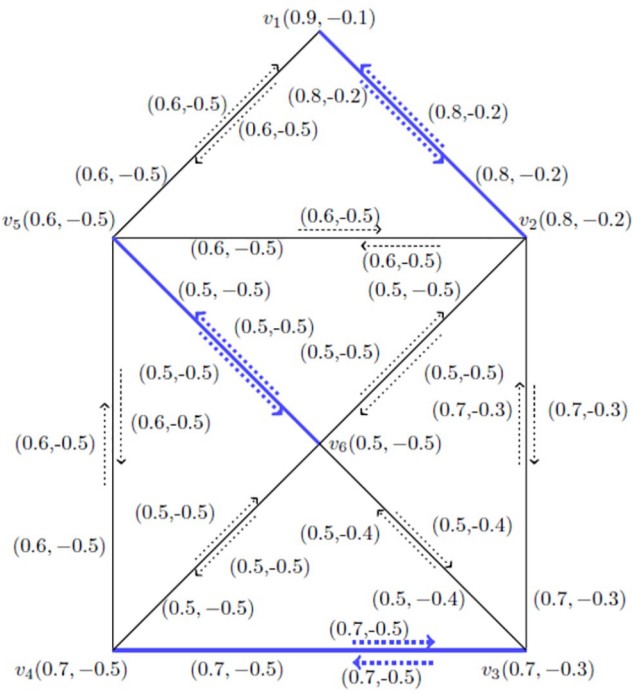

**Fig 13. An arbitrary *BFIG*.**

obtained *MMBFIN* = (1, −0.6) and got the same matching $\dot{M}_1 =$ $\langle (u_1, u_1 v_3), (u_3, u_3 v_2), (u_2, u_2 v_0) \rangle$ for *BBFIG* by using the incidence pairs. So, the result is better by using the incidence pairs as by using the vertices we have more chances of controversial issues i.e., $MMVBFIN = \frac{2.1}{2.7} * 100 = 77.78$ and by using the incidence pairs, we have $MMBFIN = \frac{0.6}{1} * 100 = 60$.

Now, there are six employees in a company. We give them membership values according to their individual performance. There edge values are defined as their work performance with other member as a group. The positive membership value of incidence pair value defines the working efficiency of two employees in a company and negative membership value defines their loss possibility in the working due to controversial issues among the employees as a group. In Fig 11 we obtained *MMVBFIN* = (4.2, −1.8) and got a matching $\dot{M} = \langle (v_2, v_2 v_1), (v_5, v_5 v_6), (v_3, v_3 v_4) \rangle$ for *BFIG* by using vertices. Now, by using Fig 13 we obtained *MMBFIN* = (2, −1.2) and got the same matching $\dot{M} = \langle (v_2, v_2 v_1), (v_5, v_5 v_6), (v_3, v_3 v_4) \rangle$ for *BFIG* by using the incidence pairs.

Both matchings either by using vertex or incidence pairs are same but incidence pairs are representing the influence of on vertex to other. So, the incidence graphs are more better as by using the incidence pairs we can see that which employee have greater influence on other or which group have better efficiency level.

## 6 Conclusion

Graph theory is very needful for presenting the data of real life problems. In this article, we enhanced the theory of *BFIGs*. The matching concept becomes very useful when it is discussed by using *BFIGs* because it also includes the controversial issues or chances of loss among the employees in a company. After introducing the concept of matching in *BFIGs*, its related

propositions, results and theorems with some examples are presented. Matching numbers are obtained to improve the working quality of the employees in a company. Finally, a decision making graph of a company is presented to reflect the working of the members and achieving maximum results by minimizing chances of loss. Our goal is to enhance this research to soft *FIGs*, q-rung *FIGs* with more theorems and applications in forthcoming articles.

## Acknowledgments

The authors are highly grateful to Editor-in-Chief and the anonymous reviewers for their valuable comments and suggestions to improve the quality of our manuscript.

## Author Contributions

**Conceptualization:** Fahad Ur Rehman, Tabasam Rashid, Muhammad Tanveer Hussain.

**Formal analysis:** Tabasam Rashid, Muhammad Tanveer Hussain.

**Investigation:** Tabasam Rashid.

**Methodology:** Fahad Ur Rehman.

**Supervision:** Muhammad Tanveer Hussain.

**Validation:** Tabasam Rashid, Muhammad Tanveer Hussain.

**Visualization:** Muhammad Tanveer Hussain.

**Writing – original draft:** Fahad Ur Rehman.

**Writing – review & editing:** Muhammad Tanveer Hussain.

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
