## [Decision Letter · Decision Letter 0]

20 Feb 2023

PONE-D-22-29739MAXIMUM MATCHING WITH APPLICATION BY USING BIPOLAR FUZZY INCIDENCE GRAPHSPLOS ONE

Dear Dr. Hussain,

Thank you for submitting your manuscript to PLOS ONE. After careful consideration, we feel that it has merit but does not fully meet PLOS ONE’s publication criteria as it currently stands. Therefore, we invite you to submit a revised version of the manuscript that addresses the points raised during the review process.

We look forward to receiving your revised manuscript.

Kind regards,

Ronnason Chinram, Ph.D.

Academic Editor

PLOS ONE

Journal Requirements:

2. "PLOS requires an ORCID iD for the corresponding author in Editorial Manager on papers submitted after December 6th, 2016. Please ensure that you have an ORCID iD and that it is validated in Editorial Manager. To do this, go to ‘Update my Information’ (in the upper left-hand corner of the main menu), and click on the Fetch/Validate link next to the ORCID field. This will take you to the ORCID site and allow you to create a new iD or authenticate a pre-existing iD in Editorial Manager. Please see the following video for instructions on linking an ORCID iD to your Editorial Manager account: " ext-link-type="uri" xlink:type="simple">https://www.youtube.com/watch?v=_xcclfuvtxQ"

Additional Editor Comments (if provided):

Your paper has been fully reviewed. Based upon my judgment and the reviewers comments, we inform you that your paper has received the major revision for the possible publication. Please see the comments of reviewers and kindly revise your paper and submit with the response letter at the earliest convenience.

Reviewers' comments:

Reviewer's Responses to Questions

**Comments to the Author**

1. Is the manuscript technically sound, and do the data support the conclusions?

Reviewer #1: Yes

Reviewer #2: Yes

2. Has the statistical analysis been performed appropriately and rigorously? 

Reviewer #1: Yes

Reviewer #2: Yes

3. Have the authors made all data underlying the findings in their manuscript fully available?

Reviewer #1: Yes

Reviewer #2: Yes

4. Is the manuscript presented in an intelligible fashion and written in standard English?

Reviewer #1: Yes

Reviewer #2: Yes

5. Review Comments to the Author

Reviewer #1: This is a nice research article. This paper should be accepted after a good revision.

Comments/Suggestions:

(1) Clearly describe the contributions in the Abstract section.

(2) An introduction should clearly highlight the motivation, problem statement, the objective of the paper, gap in the existing research and the novelty of the conducted research.

(3) English needs to be improved significantly.

(4) It would be great if you could improve the comparative analysis section.

(5) Expand literature review by citing related work. For example,

Graphs for the Analysis of Bipolar Fuzzy Information, Studies in Fuzziness and Soft Computing, Springer, DOI: 10.1007/978-981-15-8756-6, 401(2021).

(6) Change min by \\min , and max by \\max in the whole manuscript.

(7) Explain MMIBFIN problem in BBFIG more nicely.

Reviewer #2: This paper introduced a novel method to MAXIMUM MATCHING WITH APPLICATION BY USING BIPOLAR FUZZY

INCIDENCE GRAPHS.

1. COMMENT: ‘Abstract’ should be refined. There are some grammar mistakes and redundant expression which may lead to misunderstandings.

2. COMMENT: ‘Introduction’ and ‘Preliminaries’ should be shortened. Some of the contents can be merged.

3.: The example used in the paper is considered to illustrate the effectiveness of the proposed approach. However, the authors are suggested to conduct more experiments to show the effectiveness. In addition, more comparisons and analysis should be provided to prove the efficiency.

4. Please clearly describe the motivation of this paper in the Introduction Section of this paper.

5.Please clearly describe the contributions of this paper in the Abstract, the Introduction Section and the Conclusions Section of this paper.

6. Please add the following references regarding “Fuzzy graph” into the References Section of this paper and cite them in the Introduction Section of this paper:

“A Study of m−Polar Neutrosophic Graph with Applications”, Journal of Intelligent Fuzzy Systems, Journal of Intelligent Fuzzy Systems, vol. 38, no. 4, pp. 4809-4828, 2020

“A study on Regular Picture Fuzzy Graph with Applications in Communication Networks” Journal of Intelligent Fuzzy Systems,2020

6. PLOS authors have the option to publish the peer review history of their article (what does this mean?). If published, this will include your full peer review and any attached files.

Reviewer #1: No

Reviewer #2: No

---

## [Author Response · Author response to Decision Letter 0]

17 Mar 2023

Response to Reviewer

Title: Applications of maximum matching by using bipolar fuzzy incidence graphs

Decision: Major Revision

Dear reviewers thank you for the encouragement and suggestions. We have revised manuscript carefully according to the given instructions. The below are the comments and reponses of both the reviewers.

Reviewer # 1: Comments/Suggestion and their responses.

Comment 1:Clearly describe the contributions in the abstract section.

Reply: Dear reviewer we are gratified to you for your beneficial comment. We have added our contributions in the abstract. It is in highlighted form in the abstract of revised manuscript. 

Comment 2: An introduction should clearly highlight the motivation, problem statement, the objective of the paper, gap in the existing research and the novelty of the conducted research.

Reply: Thank you for your valuable suggestion. We have improved the introduction section by adding some related material and highlighted it in our revised manuscript. Now, we have added motivation, problem statement, the objective of the paper, gap in the existing research in our revised manuscript in the introductory section on page number 2 in second and third paragraphs. We have highlighted it in our revised manuscript. 

Comment 3: English needs to be improved significantly.

Reply: We are very thankful to you for your valuable suggestion. We have read the whole article carefully and corrected the mistakes and grammatical errors. Also, we have proofread the manuscript thoroughly.

Comment 4: It would be great if you could improve the comparative analysis section.

Reply:Dear reviewer we have added improved comparative analysis section in the rervised manuscript.

Comment 5:Expand literature review by citing related work. For example,Graphs for the Analysis of Bipolar Fuzzy Information, Studies in Fuzziness and Soft Computing, Springer, DOI: 10.1007/978-981-15-8756-6, 401(2021).

Reply:Thank you for the suggestion. Now, we have added the above mentioned research work in the reference section at [14].

Comment 6: min by \\min, and max by \\max in the whole manuscript.

Reply: We have changed min by \\min and \\max in our revised manuscript. 

Comment 7:Explain MMBFIN problem in BBFIG more nicely.

Reply:Thank you for the suggestion. Now, we have explained improved MMBFIN problem in BBFIG by discussing also on vertices and added more sections in section 4.

Reviewer 2: Comments/Suggestions and their responses.

 Comment 1:Abstract should be refined. There are some grammar mistakes and 

redundant expression which may lead to misunderstandings.

Reply: We are gratified to you for your beneficial comment. We have revised the abstract in our revised manuscript and corrected all grammar mistakes and removed the redundant expressions.

Comment 2: ‘Introduction’ and ‘Preliminaries’ should be shortened. Some of the contents can be merged.

Reply: Thank you for your valuable comment. We have shortened the introduction and preliminaries, with this some contenets are merged now.

Comment 3: The example used in the paper is considered to illustrate the effectiveness of the proposed approach. However, the authors are suggested to conduct more experiments to show the effectiveness. In addition, more comparisons and analysis should be provided to prove the efficiency.

Reply: Thank you for this suggestion. We have added more subsections in section 4 to illustrate the effectiveness of our proposed approach. Also we have provided more comparison and analysis to show our efficiency.

Comment 4: Please clearly describe the motivation of this paper in the Introduction Section of this paper.

Reply: Thank you for your valuable suggestion. We have improved the introduction section by adding some related material and highlighted it in our revised manuscript. Now, we have added motivation in our revised manuscript in the introductory section on page number 2 in second and third paragraphs. We have highlighted it in our revised manuscript. 

Comment 5: Please clearly describe the contributions of this paper in the Abstract, the Introduction Section and the Conclusions Section of this paper.

Reply:We are gratified to you for your beneficial comment. We have added our contributions in the abstract, and also added motivation in our revised manuscript in the introductory section on page number 2 in second and third paragraphs. It is in highlighted form in the abstract and introduction section of revised manuscript.

Comment 6: Please add the following references regarding “Fuzzy graph” into the References Section of this paper and cite them in the Introduction Section of this paper.

“A Study of m−Polar Neutrosophic Graph with Applications”, Journal of Intelligent Fuzzy Systems, Journal of Intelligent Fuzzy Systems, vol. 38, no. 4,pp.4809-4828,2020

“A study on Regular Picture Fuzzy Graph with Applications in Communication Networks” Journal of Intelligent Fuzzy Systems,2020

Reply:Thank you for the suggestion. Now, we have added the above mentioned research work in the reference section at [20] and [21] and cited them in the introduction section.

---

## [Decision Letter · Decision Letter 1]

27 Apr 2023

Applications of maximum matching by using bipolar fuzzy incidence graphs

PONE-D-22-29739R1

Dear Dr. Hussain,

We’re pleased to inform you that your manuscript has been judged scientifically suitable for publication and will be formally accepted for publication once it meets all outstanding technical requirements.

Kind regards,

Ronnason Chinram, Ph.D.

Academic Editor

PLOS ONE

Additional Editor Comments (optional):

The authors have revised the paper very carefully and addressed all the points very well, which were raised during the review process. This paper is more interesting. Hence I recommend accepting the paper for publication.

Reviewers' comments:

Reviewer's Responses to Questions

**Comments to the Author**

1. If the authors have adequately addressed your comments raised in a previous round of review and you feel that this manuscript is now acceptable for publication, you may indicate that here to bypass the “Comments to the Author” section, enter your conflict of interest statement in the “Confidential to Editor” section, and submit your "Accept" recommendation.

Reviewer #1: All comments have been addressed

2. Is the manuscript technically sound, and do the data support the conclusions?

Reviewer #1: Yes

3. Has the statistical analysis been performed appropriately and rigorously? 

Reviewer #1: Yes

4. Have the authors made all data underlying the findings in their manuscript fully available?

Reviewer #1: Yes

5. Is the manuscript presented in an intelligible fashion and written in standard English?

Reviewer #1: Yes

6. Review Comments to the Author

Reviewer #1: . Some characteristics of maximum matching 9

principal numbers in BF IG are explained which are helpful for solving the vertex and 10

incidence pair fuzzy maximization problems. Lastly, obtained maximum matching 11

principal numbers by using the matching concept to prove its applicability and 12

effectiveness for the applications in bipartite BF IG and also for the BF IG.

I accept the revised version.

7. PLOS authors have the option to publish the peer review history of their article (what does this mean?). If published, this will include your full peer review and any attached files.

Reviewer #1: No

---

## [Editor Report · Acceptance letter]

9 May 2023

PONE-D-22-29739R1 

Applications of maximum matching by using bipolar fuzzy incidence graphs 

Dear Dr. Hussain:

I'm pleased to inform you that your manuscript has been deemed suitable for publication in PLOS ONE. Congratulations! Your manuscript is now with our production department. 

Kind regards, 

on behalf of

Dr. Ronnason Chinram 

Academic Editor

PLOS ONE